An Indo-Pacific damselfish (Neopomacentrus cyanomos) in the Gulf of Mexico: origin and mode of introduction

Robertson D. Ross drr@stri.org 1
Dominguez-Dominguez Omar 2 3
Victor Benjamin 4
Simoes Nuno 5 6 7
1 Naos Marine Laboratory, Smithsonian Tropical Research Institute , Balboa , Republic of Panama
2 Laboratorio de Biologia Acuatica, Facultad de Biologia, Universidad Michoacana de San Nicolás de Hidalgo , Morelia , Michoacan , Mexico
3 Laboratorio Nacional de Análisis y Síntesis Ecológica para la Conservación de Recursos Genéticos de México, Escuela Nacional de Estudios Superiores, Unidad Morelia, Universidad Nacional Autónoma de México , Morelia , Michoacán , Mexico
4 Guy Harvey Research Institute, Nova Southeastern University , Ft Lauderdale , FL , United States of America
5 Unidad Multidisciplinaria en Docencia e Investigacion de Sisal, Facultad de Ciencias, UNAM , Sisal , Yucatan , Mexico
6 Laboratorio Nacional de Resiliencia Costera, Unidad Académica de Yucatán, Universidad Nacional Autónoma de México , Sisal , Yucatán , Mexico
7 Harte Institute for Gulf of Mexico Studies, Texas A&M University—Corpus Christi , Corpus Christi , TX , United States of America
Kramer Donald
Electronic publication date: 2018 Feb 7
Publication date: 2018
Volume: 6
Electronic Location ID: e4328
Received 2017 Nov 8; Accepted 2018 Jan 15
Copyright: ©2018 Robertson et al.
Copyright year: 2018
Copyright holder: Robertson et al.
License: This is an open access article distributed under the terms of the Creative Commons Attribution License, which permits unrestricted use, distribution, reproduction and adaptation in any medium and for any purpose provided that it is properly attributed. For attribution, the original author(s), title, publication source (PeerJ) and either DOI or URL of the article must be cited.
License URL: https://creativecommons.org/licenses/by/4.0/

Keywords: Invasive fish, Greater caribbean, Offshore petro-platform, DNA barcode, Coral-reef fish

Funding: International Barcode of Life Project 2008-OGI-ICI-03 Harte Charitable Foundation to the Fundación de la Biodiversidad Marina de Yucatan Cordinación de la Investigación Cinetifica-UMSNH Barcoding at the Center for Biodiversity Genomics was supported by the International Barcode of Life Project (http://iBOL.org) with funding from the Government of Canada via the Canadian Centre for DNA Barcoding, as well as from the Ontario Genomics Institute (2008-OGI-ICI-03), Genome Canada, the Ontario Ministry of Economic Development and Innovation, and the Natural Sciences and Engineering Research Council of Canada. Fieldwork in the SWGoMx was supported by a grant from the Harte Charitable Foundation to the Fundación de la Biodiversidad Marina de Yucatan. Laboratory work at UMSNH was supported by Cordinación de la Investigación Cinetifica-UMSNH. The funders had no role in study design, data collection and analysis, decision to publish, or preparation of the manuscript.

==============================
The Indo-West Pacific (IWP) coral-reef damselfish Neopomacentrus cyanomos is well established across the south-west Gulf of Mexico (SwGoMx). Comparisons of mtDNA sequences of the SwGoMx population with those from conspecifics from 16 sites scattered across its native geographic range show that the SwGoMx population is derived from two of four native lineages: one from the north-west Pacific Ocean, the other from the northern Indian Ocean. Three hypotheses address how this species was introduced to the SwGoMX: (1) aquarium release; (2) borne by cargo-ship; and (3) carried by offshore petroleum platform (petro-platform). The first is unlikely because this species rarely features in the aquarium trade, and “N. cyanomos” traded to the USA from the sole IWP source we are aware of are a misidentified congener, N. taeniurus. The second hypothesis is unlikely because shipping has not been associated with the introduction of alien damselfishes, there is little international shipping between the IWP and the SwGoMx, and voyages between those areas would be lengthy and along environmentally unfavorable routes. Various lines of evidence support the third hypothesis: (i) bio-fouled petro-platforms represent artificial reefs that can sustain large and diverse populations of tropical reef-fishes, including N. cyanomos in the SwGoMx; (ii) relocation of such platforms has been implicated in trans-oceanic introductions leading to establishment of non-native populations of such fishes; and (iii) genetic characteristics of the SwGoMx population indicate that it was established by a large and diverse group of founders drawn from the IWP regions where many petro-platforms currently in the SwGoMx and other Atlantic offshore oilfields originated.

Introduction

The damselfish Neopomacentrus cyanomos (Bleeker,1856), which is native to coral reefs of the tropical Indo-West Pacific (IWP) (Allen, 1991), was first recorded in the south-west Gulf of Mexico (SwGoMx) in mid-2013 (González-Gándara & De la Cruz-Francisco, 2014). However, by then it was common on both coastal and offshore reefs dispersed across at least 350 km of that area (Robertson et al., 2016a). Since then, this species has been found across most of the SwGoMx, and, most recently, on petro-platforms, artificial- and natural reefs in both the north-west and north-east Gulf of Mexico (Schofield, 2017). After the lionfish (Pterois volitans), which arguably represents a single species in the west Atlantic (cf. Wilcox et al., 2017), N. cyanomos is only the second species of IWP coral-reef fish to have established a substantial population across a large area of the tropical Greater Caribbean. This paper deals with two questions: (i) from where in its native range did the SwGoMx population of N. cyanomos originate, and (ii) by what means were the fish that established that population transported to the Gulf of Mexico.

To address these two questions we compared DNA sequences of the mitochondrial DNA cytochrome c oxidase I gene (CO1 sequences) to assess relationships among three groups of fish: (i) aquarium-trade fish purchased in the USA that were supposedly “N. cyanomos” shipped from the Philippines; (ii) a large group of individuals of N. cyanomos we collected from two reefs in the SwGoMx; and (iii) samples of N. cyanomos collected at 16 sites scattered across most of its native range in the IWP.

We used these genetic data to assess the following predictions relating to three hypothesized modes by which tropical reef-fishes have been transported long distances to sites well outside their native ranges: (1) release of aquarium specimens (Semmens et al., 2004; Schofield, Morris Jr & Akins, 2009); (2) carriage by cargo-ship, on the hull or in ballast water (González-Gándara & De la Cruz-Francisco, 2014); and (3) carriage by marine platforms used for exploration and production of offshore oil and gas (“petro-platforms”) (Robertson et al., 2016b). We predicted that if the N. cyanomos population in the SwGoMx was established by aquarium release, then individuals from the SwGoMx should match genetically to those supplied by the aquarium trade from the Philippines (apparently the sole source of fish labelled as this species for the US trade to the Americas during the early 2000s, the only period for which published data are available; https://www.aquariumtradedata.org/, accessed July 15, 2017). Since it is likely that an aquarium release of an obscure species such as N. cyanomos, a minor component in the international fish trade, would involve a relatively small number of individuals, then the N. cyanomos population in the SwGoMx should have low genetic diversity and show evidence of having passed through a genetic bottleneck due to its expansion from a small number of founders (a founder effect). Similarly, if the introduction were from transport by cargo-ship, a method not conducive to transport of large numbers of individual fishes in a single event, we predicted a SwGoMx population with genetic characteristics essentially the same as those for a population established by aquarium release, i.e., low genetic diversity, with evidence of a founder effect (Bernardi, Golani & Azzurro, 2010; Seixas et al., 2017). In contrast, we predicted that transfer by petro-platforms would produce a markedly different genetic profile in the GoMx population. These platforms can host substantial populations of a range of species of tropical reef-fishes (Hastings, Ogren & Mabry, 1976; Ferreira, Gonçalves & Coutinho, 2006; Friedlander et al., 2014; Pradella et al., 2014), and at least one small platform in the SwGoMx supports a high-density population of many tens of thousands of N. cyanomos (Simoes & Robertson, 2016). Hence, a platform being transported from the IWP potentially could carry a substantial number of individuals representing a cross section of the population(s) of N. cyanomos from the IWP site(s) where it had been constructed or stationed before arriving in the GoMx. A GoMx population that was genetically diverse and showed no signs of having passed through a genetic bottleneck due to small founder-population size would be consistent with such a mode of introduction. In addition, that GoMx population should be most closely related to an IWP population(s) in an area(s) that contain offshore oilfields and where petro-platforms were constructed or stationed before being moved to the SwGoMx.

Materials and Methods

Research at Cayo Arcas in 2016 was permitted by the Mexican Navy (SEMAR DO: 112/162), which also provided logistical support for that activity. An ACUC permit from the Smithsonian Tropical Research Institute (no. 2017-1107-2020) approved methods used to collect specimens of Neopomacentrus cyanomos there.

Study species and origin of DNA sequences

The native IWP range of N. cyanomos extends from the Red Sea and the Persian Gulf to south-east Africa, and eastward to Japan, the Philippines, the Solomon Islands and New Caledonia in the West-Pacific. (Allen, 1991; https://www.gbif.org/species/2398535, accessed Sept. 20, 2017). The range of N. cyanomos encompasses most of the range of the entire genus, which extends only further southeast, to the south-central Pacific (Allen, 1991; https://www.gbif.org/species/2398526, accessed Sept. 20, 2017). No members of the genus occur naturally in the Atlantic Ocean (Allen, 1991), and none are yet known from the Mediterranean Sea, either naturally (Allen, 1991) or as a result of introduction (Arndt & Schembri, 2015).

The DNA sequences of N. cyanomos that we analysed came from four sources: (i) tissue samples of fish we collected in the Gulf of Mexico; (ii) tissue samples from fish collected at various sites in the IWP that we obtained from other researchers; (iii) sequences independently obtained by other researchers from different sites in the IWP and compiled by the Fish Barcode of Life Initiative (FISH-BOL; http://www.fishbol.org/); and (iv) tissue samples from putative aquarium-trade N. cyanomos.

The 128 sequences we analysed included 54 from documented N. cyanomos (many identified by GR Allen), that had been collected by a variety of researchers, including contributors to the FISH-BOL database, at 16 sites (1–8 fish per site) scattered across most of the native IWP range. These sequences were compared to sequences of 65 individuals of N. cyanomos we collected from the non-native population in the SwGoMx on Campeche Bank, two from Madagascar reef in 2015 (Robertson et al., 2016b) and 63 from Cayo Arcas reef in 2016. After collection using the anaesthetic clove-oil those 65 fish were euthanized with an overdose of that oil, and a pectoral fin from each was preserved in 95% ethanol for use as a tissue sample from which we obtained DNA. In addition, we analysed sequences from nine aquarium-trade fish we purchased in the USA that were identified by the US wholesale dealers as “N. cyanomos” from the Philippines (eight from a single group purchased in Los Angeles, California, and one from New York, New York). Table S1 provides information on sample sites, numbers of sequences and their Genbank Accession numbers, and indicates how we obtained all DNA sequences used in this study, both those from Fish we collected or tissue samples sent to us by third parties, and those from other contributors to the FISH-BOL database.

DNA sequencing

DNA extractions were performed with the NucleoSpin96 (Machery-Nagel, Easton, PA, USA) kit according to manufacturer’s specifications under automation with a Biomek NX liquid-handling station (Beckman-Coulter, Brea, CA, USA) equipped with a filtration manifold. PCR amplifications were performed on volumes of 12.5 µl compose of 6.25 µl of 10% trehalose, 2 µl of ultra pure water, 1.25 µl of 10 × PCR buffer (10 mM KCl, 10 mM (NH4)2SO4, 20 mM Tris-HCl (pH8.8), 2 mM MgSO4, 0.1% Triton X-100), 0.625 µl of MgCl2 (50 mM), 0.125 µl of each primer (0.01 mM), 0.0625 µl of each dNTP (10 mM), 0.0625 µl of Taq DNA polymerase (New England Biolabs, Ipswich, MA, USA), and 2 µl of template DNA. The PCR conditions consisted of 94 °C for two min, 35 cycles of 94 °C for 30 s, 52 °C for 40 s, and 72 °C for one min, with a final extension at 72 °C for 10 min. Sequences from the native-range specimens (other than those available on FISH-BOL) were obtained from the Biodiversity Institute of Ontario, University of Guelph, Canada, while those from the SwGoMx were provided by Macrogen Inc., Amsterdam, The Netherlands.

Phylogenetic analysis and haplotype network reconstruction

Nucleotide sequences were edited and manually aligned in Mega v.6.06 (Tamura et al., 2013) and examined with a chromatogram. A final alignment of 594 base pairs was used in the analyses. Maximum likelihood (ML) analysis was conducted with RAxMLGUI v. 1.3.1 (Stamatakis, 2006; Silvestro & Michalak, 2012), performing 10,000 bootstrap replicates, and using the evolutionary substitution model estimated with the GTR+G model, as obtained in jModelTest (Posada, 2008) according to AIC (Posada, 2008). The relative stability of clades was evaluated by 1,000 non-parametric bootstrap replicates. N. azysron and N. sororius (Genbank # KP194962.1, and BOLD # PCOM166-10, respectively) were used as out-groups. The haplotype network was constructed using PopART (available at http://popart.otago.ac.nz/downloads.shtml) and applying the Median-Joining method (Bandelt, Forster & Röhl, 2000).

Historical demography

To detect signatures of demographic changes in populations of each of the four native lineages and the two lineages of N. cyanomos found in the SwGoMx (those belonging to groups 1 and 4 in the haplotype network of Figs. 1 and 2), Tajima’s D (Tajima, 1989), and Fu’s Fs (Fu & Li, 1993) neutrality tests were calculated, in DNAsp v5 (Librado & Rozas, 2009). Evidence of an expanding population of a lineage would be assumed if significant negative values of Tajima’s D and Fu’s F-statistics were obtained (Mousset, Derome & Veuille, 2004). Additionally, a pairwise mismatch distribution was computed for each of those six populations in DNAsp v5 (Librado & Rozas, 2009). Populations that have been historically stable are predicted to have multimodal mismatch distributions, whereas those that have undergone a recent expansion are predicted to be unimodal (Slatkin & Hudson, 1991). To test if the observed mismatch distributions predicted a stable or an expanding population model, differences between observed and expected (multimodal) distributions were evaluated using the Harpending’s raggedness index (Harpending et al., 1993) and the R2 statistic (Ramos-Onsins & Rozas, 2002). Diversity indices were calculated for each genetic group among both the native-range and SwGoMx populations, and for combined populations, using the program DNAsp v5 (Librado & Rozas, 2009).

Results

The haplotype network for the entire set of 128 sequences is presented in Fig. 1, and the locations from which individuals in the four major haplotype groups of N. cyanomos identified in that figure were obtained are shown in Fig. 2. Genetic diversity parameters of these different haplotype groups are displayed in Table 1.

Figure 1 Haplotype network.

Haplotype network of N. cyanomos from its native range and the southwest Gulf of Mexico, and of aquarium-trade “N. cyanomos” (= N. taeniurus) from the Philippines. Number of mutational steps: each red line indicates a single step, boxed number indicates number of multiple steps. Each multicolored haplotype pie indicates the abundance of a haplotype found at multiple locations.

The aquarium-trade specimens supposedly from the Philippines are only distantly related to all N. cyanomos sequenced here, and were identified by us, both morphologically and genetically, as a different species, Neopomacentrus taeniurus (Bleeker, 1856), which is native to brackish non-reef environments and widely distributed in the IWP (Allen, 1991). The mtDNA lineage of those aquarium-trade fish is separated by 80-89 mutation steps from all four haplotype groups of N. cyanomos, from both the native sites and the SwGoMx (Fig. 1).

Figure 2 Locations of sites with different haplotype groups.

Locations of sites within the Indo-west Pacific and the southwest Gulf of Mexico from which individuals were obtained from four different haplotype groups (see Fig. 1) of N. cyanomos, and of aquarium-trade “N. cyanomos” (= N. taeniurus) from the Philippines. The ML (bootstrap support >85% in all cases) tree is shown as an inset. The color scheme illustrates the geographic distributions of the four major genetic lineages of N.cyanomos.

Table 1 Genetic diversity parameters for different genetic groups of N. cyanomos from both the native Indo-West Pacific range and the Gulf of Mexico.

Genetic group	N	S	H	π	h	D	Fs	R2	r	
Group 1 native	13	8	8	0.0039	0.910	−0.40	−3.11	0.11	0.06	
Group 2 native	9	5	4	0.0024	0.583	−0.91	−0.29	0.12	0.25	
Group 3 native	29	8	7	0.0027	0.700	−0.66	−1.22	0.09	0.19	
Group 4 native	5	4	4	0.0030	0.900	−0.41	−1.20	0.21	0.13	
All native	54	43	23	0.0173	0.922	–	–	–	–	
Group 1 GoMx	38	5	5	0.0028	0.683	−1.00	1.05	0.08	0.16	
Group 4 GoMx	26	7	6	0.0020	0.603	−1.08	−1.93	0.08	0.06	
All GoMx	6	5	27	11	0.0183	0.823	–	–	–	
All native + GoMx	121	49	30	0.0195	0.919	–	–	–	–	
Group 1 native + GoMx	51	12	11	0.0036	0.791	−0.57	−2.39	0.08	0.04	
Group 4 native + GoMx	32	8	8	0.0022	0.635	−1.02	−2.70	0.08	0.05	
Notes.

N sample size

S no. polymorphic sites

H no. haplotypes

π nucleotide diversity

h haplotype diversity

D Tajima’s D

Fs Fu’s Fs statistic (all values of D and Fs were non-significant at p > 0.05)

R2 Ramos-Onsins & Rozas’ R2 statistic

r Harpending’s raggedness index (all values of R2 and r were non-significant at p > 0.05)

Native Indo-West Pacific native range

GoMx Gulf of Mexico

All native-range specimens of N. cyanomos belonged to four closely related haplotype groups and all SwGoMx fish belonged to two of those haplotype groups (groups 1 & 4; Fig. 1.). The native-range specimens had a total of 23 haplotypes, 12 of those from groups 1 and 4 (18 of 54 individuals; Fig. 1, Table 1). The 65 SwGoMx sequences included 11 haplotypes, four of which were shared with native-range specimens of groups 1 and 4 (Fig. 1, Table 1). Haplotype diversity and nucleotide diversity were high within each of the four genetic groups from the native-range (Table 1). While haplotype diversity also was high in the SwGoMx populations of groups 1 and 4, in each case the level was lower than in the native-range population of the same group (Table 1). Nucleotide diversity in each of groups 1 and 4 of the SwGoMx population was a little lower than in the native group of each case, but similar to nucleotide diversity in the other two groups (2 and 3) from the native population (Table 1).

The SwGoMx population comprised roughly equal numbers of individuals drawn from the two haplotype groups that are largely geographically separated in the native range, with group 1 restricted to the northern Indian Ocean and group 4 found in samples from Indonesia and Taiwan in the north-west Pacific Ocean (Fig. 2). The only exception to this pattern of geographic separation is one individual of group 4 that was collected at Gujarat, in India, at the same location as two individuals of group 1.

The results of the neutrality tests showed that Tajima’s D and Fu’s Fs values were not statistically significant for any of the four groups in the native range, or for either of groups 1 and 4 in the SwGoMx (Table 1). Each of those six groups showed a multimodal mismatch distribution, with non-significant results for both Harpending’s raggedness index and the R2 statistic (Table 1). Thus, none of those six groups showed evidence of having passed through a bottleneck due to a founder effect.

Discussion

The analysis of mtDNA sequences indicate that the SwGoMx population of N. cyanomos is unrelated to the aquarium-trade fish from the Philippines, which represent a different species, N. taeniurus. Most significantly, the SwGoMx population is derived from two distinct genetic lineages of N. cyanomos found in the native range, is genetically diverse, and neither of those GoMx lineages shows signs of having passed through a genetic bottleneck due to a founder effect. Lastly, the SwGoMx population is derived from a combination of a lineage found in the northern Indian Ocean and another from two widely separated locations in the northwest Pacific Ocean (Taiwan and Indonesia). These results permit an evaluation of alternative hypotheses on the origin and mode of introduction of this alien species into the Gulf of Mexico.

Aquarium-release hypothesis

The aquarium-trade specimens sold as “N. cyanomos” proved to belong to a different species. As part of a separate phylogenetic study of the genus, one of us (B Victor) has obtained mtDNA CO1 sequences of specimens of all 15 nominal species of Neopomacentrus that were collected in the field and identified by an expert in damselfish taxonomy (GR Allen). The mtDNA sequences of the Philippines aquarium-trade specimens are nested within a cluster of barcodes of specimens identified as N. taeniurus that were obtained from multiple locations (sequences in Appendix I). In addition, our morphological examinations confirmed this identification: Species of Neopomacentrus fall into two morphological groups, one (including N. cyanomos) with the lower edge of the suborbital bone hidden under a covering of scales, and the other (including N. taeniurus) in which the suborbital margin is exposed and not covered by scales (Allen, 1991). Specimens from the SwGoMx we examined all have a hidden suborbital margin, while the aquarium-trade specimens of “N. cyanomos” we examined all have exposed suborbitals. N taeniurus has a similar color pattern to, and can be easily mistaken for N. cyanomos.

Aquarium-trade data presented by Rhyne et al. (2015) indicate that “N. cyanomos”, which is not brightly colored like many desirable aquarium species, is a very small component of the aquarium trade in IWP reef-fishes to the USA (not sold in sufficient numbers to be represented among the top 20 species of damselfishes), and that the Philippines apparently is the sole native-range source for that trade to the USA. Neopomacentrus cyanomos is known to occur in the Philippines, but whether or not any aquarium-trade imports from there actually are that species remains uncertain. Furthermore, it is not clear which mitochondrial lineage(s) of N. cyanomos is present in the Philippines, given the proximity of that archipelago to not only of the Group 4 lineage (one of the SwGoMx lineages), but also the Group 2 lineage at Western Papua (Fig. 2). Even if the Group 4 lineage of N. cyanomos that is present in the SwGoMx is exported from the Philippines in the aquarium trade to the US, the northern Indian Ocean lineage (Group 1) that forms a major part of the SwGoMx population would not be expected to be part of the trade.

While keeping marine aquarium-fish is popular and widespread in countries like the US, that is not the case in Mexico (N Simoes, personal observations), where attention to the international trade in marine aquarium fish is focused primarily on export of native species (e.g., Lango Reynoso et al., 2012). Any individuals of alien marine fishes obtained by aquarists in Mexico likely would have been derived from US wholesalers, the main global recipient in the aquarium trade. Personnel at a large public aquarium at the city of Veracruz, in the SwGoMx, the only such installation in the Mexican part of the Gulf of Mexico, have confirmed that N. cyamonos has never been maintained in that facility (H Pérez-España, pers. comm., 2017). Almost all recorded aquarium-trade releases of exotic marine fish in or adjacent to the GoMx have occurred in southern Florida, where at least 31 species (14 different families) of alien reef-fishes have been recorded (Semmens et al., 2004; Schofield, Morris Jr & Akins, 2009). N. cyanomos would be expected to have established in southern Florida first if its GoMx population was derived from aquarium release. It has not been discovered there (Schofield, 2017; accessed Dec. 20, 2017), despite the fact that southeast Florida hosts an abundance of divers, and is intensively monitored by the REEF organization (http://www.reef.org) for sightings of exotic species. REEF was made aware of the presence of N. cyanomos in the Gulf of Mexico by the publication of González-Gándara & De la Cruz-Francisco (2014), and members of that organization have been looking for this species on south Florida reefs.

The genetic data do not support the aquarium-release hypothesis. Both SwGoMx lineages of N. cyanomos are genetically diverse, and the results of both the neutrality tests and mismatch distributions indicate a stable population that did not pass through a genetic bottleneck due to a founder effect, and hence was not established by a few fish released from an aquarium.

Cargo-ship carriage hypothesis

There are very few instances of long-distance introductions by alien damselfishes. One of those evidently is due to a range expansion within the Pacific by a species that associates with flotsam (Abudefduf vaigiensis (Quoy & Gaimard, 1825), Coleman et al., 2014). Another introduction (of another Abudefduf species that also associates with flotsam) from the Atlantic to the Mediterranean, has been attributed to ship-transport by Occhipinti-Ambrogi et al. (2011), without further detail or presentation of any justification (see Tsadok et al., 2015 for resolution of the identity of this species; and Pajuelo et al., 2016, below).

Ship-transport was proposed as the means of introduction of N. cyanomos into the SwGoMx by González-Gándara & De la Cruz-Francisco (2014). Movements of cargo-ships in the normal course of business have been implicated in the long-distance transport of exotic marine fishes well beyond their native ranges, usually in ballast water (Wonham, Carlton & Ruiz, 2000), but, potentially, also in sea-chests on the outside of the hulls of ships. Damselfishes are not included among the species thought to be introduced by ballast water (Wonham, Carlton & Ruiz, 2000) and none were found on the hulls of cargo ships examined in Brazil by Ferreira, Gonçalves & Coutinho (2006). Sea chests are screened containers on the outside of the hulls that house the water intakes of a ship, inside which diminutive, crevice-dwelling reef-fishes such as gobies and blennies have been found (Wonham, Carlton & Ruiz, 2000; Coutts & Dodgshun, 2007). However, N. cyanomos is not a crevice-living species and typically feeds on plankton in a flow of water 0.5–1 m above the substrate rather than maintaining contact with it. Whether a species with these characteristics is likely to be able to survive a long voyage in a sea-chest with functioning water intakes, and do so in large enough numbers to establish a genetically diverse population when it arrives in a different ocean is far from certain.

A modern cargo-ship, which typically will not have a heavily fouled hull, travelling at 20–25 knots (37–46 kph) on a voyage from the western Pacific (Hong Kong) to the SwGoMx via the Panama Canal would take ∼19–23 days. A similar voyage would take ∼16–20 days from Northwest India via the Suez Canal (both calculated on http://sea-distances.org, accessed Sept. 20 2017), not counting stoppages at any ports during the voyage. Survivorship in ballast water is time-dependent for many marine organisms (Muirhead et al., 2015) and, in addition to the lengthy duration of a transit by either route, the Pacific route would pose additional environmental barriers for transport of tropical reef-fish hitching a ride on the hull of a cargo-ship going to the Gulf of Mexico via that route. Most such traffic goes via California (Fig. S1), which has a temperate climate with cold winter waters. However, average water temperatures near the major port of Long Beach, California, peak at ∼20 °C for several months during summer, the same as average minimum winter temperatures at the latitudinal limits of the IWP range of N. cyanomos. In addition, sea temperatures off California are elevated above the norm during El Niño events. Thus, the sea temperature regime on the coast of California would reduce, but not eliminate, the possibility of successful carriage to Mexico. Second, passing through the Panama Canal (see Fig. S1) exposes any hull-associated organisms to the freshwater lake that constitutes most of that canal, and represents an effective barrier to transfer of stenohaline reef-fishes between the oceans separated by the Isthmus of Panama (although a few euryhaline, non-reef fishes that live in brackish environments have made the transit; reviewed in Cohen, 2006; Robertson & Allen, 2015; Robertson & Van Tassell, 2015).

In addition, due to the fact that the Panama Canal represents a major global choke point for international shipping (see Fig. S1), one would expect the first occurrence of marine reef-fishes transported across the Pacific in this manner to be on reefs adjacent to the Pacific end of the canal, where ships often stop before making a canal transit. However, there are no records of such occurrences (review in Cohen, 2006; Robertson & Allen, 2015; Robertson & Van Tassell, 2015), at either end of the canal.

As with the aquarium-release hypothesis, the genetic data do not support this hypothesis, since both the SwGoMx lineages of N. cyanomos are genetically diverse, and show no signs of having passed through a genetic bottleneck and hence was not established by a few founders, i.e., by a few fish transported in ballast water or on the hull of a cargo ship.

If N. cyanomos is capable of being carried far from its native range by cargo-ship in large enough numbers to establish a genetically diverse population, that would be much more likely to occur near major ports (e.g., Miami, Florida, or Panama, or somewhere in the Mediterranean), than in the southwest Gulf of Mexico. However, there are no indications that such has happened.

Petro-platform translocation hypothesis

Offshore petro-platforms have been shown to support an abundance of reef- fishes belonging to a broad variety of taxa, in the Gulf of Mexico (Hastings, Ogren & Mabry, 1976; Topolski & Szedlmayer, 2004; Ajemian et al., 2015), Brazil (Ferreira, Gonçalves & Coutinho, 2006), California (Claisse et al., 2014; Martin & Lowe, 2010), the Mediterranean (Consoli et al., 2013), west Africa (Friedlander et al., 2014), Australia (Pradella et al., 2014), the Canary Islands (Pajuelo et al., 2016), and the Persian Gulf (Torquato et al., 2017). In the SwGoMx, we have observed N. cyanomos living on two such platforms, including in superabundance on a relatively lightly-fouled platform near Cayo Arcas, in the southwest corner of Campeche Bank (up to ∼100 individuals m−2; Simoes & Robertson, 2016). That platform is one of several hundred platforms in the area of the greatest concentration of such structures in the SwGoMx (https://portal.cnih.cnh.gob.mx/iicnih/?lng=en_US, accessed Sept. 20, 2017).

Translocated petro-platforms have long been known to be vectors of long-distance transport of sessile fouling organisms (Foster & Willan, 1979; Bax et al., 2003; Hopkins & Forrest, 2010; Wanless et al., 2010; Yeo et al., 2010; Mineur et al., 2012). Recently, such movements have been implicated in the transoceanic transport of western Atlantic reef-fishes to the Canary Islands, where a platform-servicing center has been established (Falcon et al., 2015; Pajuelo et al., 2016), as well as to the Mediterranean (Dulcic & Dragicevic, 2013; Pajuelo et al., 2016). Some reef-fish species apparently carried to the Canary Islands by petro-platforms have established breeding populations there (Falcon et al., 2015; Pajuelo et al., 2016). Petro-platforms may help exotic reef-fishes establish off the coast of west Africa (Friedlander et al., 2014), and translocated platforms may have carried eastern Atlantic reef-fishes to Brazil, where some have recently become established (Pajuelo et al., 2016; Anderson et al., 2017).

Relocated petro-platforms originating in the Indian Ocean are thought to have transported alien fishes to the Mediterranean Sea (Galil, 2008). Among the exotic reef-fishes newly recorded in the Canary Islands, one species identified by Falcon et al. (2015; plate 4B) as the (naturally trans-Atlantic) damselfish, Abudefduf taurus (Muller & Troschel, 1848), actually is most likely the IWP species A. sordidus (Forsskal, 1775) (GR Allen, pers. comm., 2016). Since A. sordidus is a dull-colored species that is not found in the aquarium trade (Rhyne et al., 2015), this occurrence represents support for the hypothesis that IWP reef-fishes are carried from their native ranges into the Atlantic by petro-platforms. Another example, in the Pacific Ocean, reported by Myers & Donaldson (1996) and Myers (1999, p 183), is the damselfish Neopomacentrus violascens (Bleeker, 1848), which they proposed was transported from the Philippines to Guam, where it is not native, by a bio-fouled military barge. It has now established a population at Guam, living on bio-fouled mooring chains rather than on natural substrata.

Are there movements to the SwGoMx of petro-platforms from the two areas of the native range of N. cyanomos that contain the lineages of that species found in the SwGoMx? The website http://www.infield.com/rigs (accessed Dec. 20, 2017) provides information on the construction sites of drilling-platforms and drill ships currently located in the offshore oilfields of different countries. A review of information on that site of the origins of such platforms and ships showed that about 60% of those listed as currently being used in the SwGoMx were constructed at sites within the native ranges of the two lineages found in the SwGoMx. Many drilling rigs currently in use in other offshore oilfields in the tropical Atlantic also were constructed at sites within the native range of N. cyanomos (Table 2). In some cases rigs were moved from those construction sites directly to the Atlantic oilfield; in other cases, particularly for older rigs, their movements prior to arrival at those offshore fields are unclear. These linkages produced by relocations of drilling-rigs represent only part of a global web of movements of new and used drilling-rigs, drill-ships, support vessels, and parts of production platforms within and between oceans that could result in establishment of exotic sessile and mobile reef organisms well beyond their native ranges (Wanless et al., 2010; Yeo et al., 2010; Pajuelo et al., 2016).

Table 2 Indo-West Pacific sources of drilling-rigs present in late 2017 on offshore oilfields of the tropical Atlantic Ocean.

		Construction site	
Atlantic oilfields	No. rigs	Persian gulf	Malaysia	Singapore	China	Taiwan	South Korea	Japan	
Brazil	13	Yes	–	Yes	Yes	–	Yes	–	
Colombia	1	Yes	–	–	–	–	–	–	
Mexico	30	Yes	–	Yes	Yes	Yes	Yes	–	
Trinidad	1	–	–	Yes	–	–	–	–	
USA	17	–	–	Yes	Yes	Yes	–	–	
West Africa	13	Yes	Yes	Yes	–	–	–	Yes	
Notes.

1 West Africa = Angola, Cameroon, Congo, Ghana, Ivory Coast and Nigeria.

2 Yes indicates rig(s) at an Atlantic site originated from the particular construction site.

3 No. rigs = no. rigs present at an Atlantic site that were constructed at one or more Indo-West Pacific sites.

4 Venezuela is not included because no rigs of Indo-West Pacific origin are currently based there.

5 Source: http://www.infield.com/rigs/, accessed December 20, 2017. Data from this website only cover currently present rigs, and do not include rigs no longer at a site; e.g., the semi-submersible drilling-rig Deepwater Horizon, which exploded and sank off Louisiana in 2010, is not included among the list of rigs for the US section of the Gulf of Mexico.

Drilling-rigs are moved long distances either by being towed by tugboats while in the sea (a wet-tow), or by being transported out of the water on the deck of a heavy-lift ship (a dry-tow). Although dry-tows are more commonly used for long-distance and interoceanic relocations of rigs, both dry- and wet-tows have been used to move rigs from the IWP to the GoMx around the southern tip of Africa (e.g., a wet-tow: Mb50’s “Liquid Mud” Rant, 2013; a dry-tow: Rigzone, 2004). Wet-towed rigs moved by this route would have the potential to transport marine organisms from the IWP to the GoMx or to any other Atlantic offshore oil-field.

The southern tip of Africa has a temperate environment that very likely inhibits the transport of tropical fish from the Indian Ocean to the Atlantic Ocean. However, summer temperatures at that tip are ∼20  °C, similar to winter temperatures at the latitudinal limits of the native geographic range of N. cyanomos. In addition, temperatures there are distinctly warmer during El Niño years (Dufois & Rouault, 2012), and there are periodic leakages of warm water from the southern Indian Ocean around the southern tip of Africa into the South Atlantic (Duncombe Rae, 1991). Thus temperatures suitable for the rig-transport of N. cyanomos are intermittently available off that tip.

Whether any of the rigs employed in the GoMx arrived there from the IWP via the Suez Canal is unclear. Limitations in the capacity of that canal may oblige some rigs to move between the Mediterranean and IWP via southern Africa (e.g., Offshore Energy Today, 2011). However, rigs have been wet-towed from the Persian Gulf to northern Europe via the Suez canal (e.g., Rigzone, 2008). Sea temperatures are at or above 20 °C for 6–8 months each year in the southern Mediterranean, which would facilitate transport of tropical organisms on rigs passing through. However, the only connections between the IWP and tropical Atlantic oilfields that we were able to find through web searches involved relocations of rigs around southern Africa.

The genetic structure of the SwGoMx population of N. cyanomos indicates it was established by two large and genetically diverse groups of founders, which did not go through a genetic bottleneck during the expansion of their populations from a small group of founders in that area. These results support the hypothesis that a large number of fish were transported to that area, which would be most consistent with the movement of one or more petro-platforms from the native range. Neopomacentrus cyanomos appears to be the first known tropical reef-fish to have successfully established outside its native geographic range, in another ocean, as a result of transport by a petro-platform.

How many introductions of N. cyanomos to the GoMx?

The fact that the SwGoMx population of N. cyanomos is derived from two lineages mostly confined to different oceans raises the question of whether that population was derived from two well-separated sources and two separate introductions. While those lineages are largely allopatric, both are present at Gujarat, in northwestern India (Fig. 2). That occurrence and the absence of the Pacific lineage (group 4; Fig. 2) at Sri Lanka (in an admittedly small sample of fish), which is closer to the Pacific than is Gujarat, suggests that the presence of group 4 at western India may not be natural. Is there evidence that offshore oil-industry activity provides connections that could have carried group 4 fish from the Pacific to Gujarat?

There is a major, active offshore-petroleum area immediately to the south of the broad Kathiawar peninsula of Gujarat that projects well into the Indian Ocean (e.g., https://watchers.news/data/uploads/2012/04/Untitled1.png, accessed Sept. 15, 2017). Of 41 petro-platforms used in India that are listed by http://www.infield.com/rigs, (accessed July 15, 2017), 15 were constructed in Singapore and two in southeast China, near the Taiwan sample of the Group 4 lineage of N. cyanomos. Further, there is a major global shipbreaking site on either side of the Gujarat peninsula, one at Gaddani in Pakistan, ∼360 km west of Gujarat, the other at Alang, at the eastern base of that peninsula, in India. A list of decommissioned vessels of various types that were sent to those two sites during 2015–2016 (provided by http://www.shipbreakingplatform.org) shows that decommissioned offshore platforms and support vessels do get taken from various locations within the northwest Pacific (Indonesia, Singapore, Vietnam, south-east China) to both those shipbreaking yards. In addition, other types of decommissioned vessels also make similar last voyages. Decommissioned vessels transiting to a shipbreaker do not have clean hulls; rather, typically they are heavily bio-fouled (Davidson et al., 2008), which would facilitate long-distance transport of fishes such as N. cyanomos.

Thus, there are interoceanic connections involving movements of drilling-platforms, drill-ships, offshore support vessels, and decommissioned vessels of various types that could be responsible for the occurrence of both lineages in the sample from Gujarat. The existence of such connections supports the hypothesis that the occurrence of N. cyanomos in the GoMx could well be due to a single introduction (the simplest explanation) rather than multiple introductions from different source locations.

When and where in the GoMx was N. cyanomos initially introduced?

Neopomacentrus cyanomos was first recorded at Coatzacoalcos, in the southwest corner of the GoMx in July 2013 by González-Gándara & De la Cruz-Francisco (2014), who thought it had been introduced directly to that port from its native range. However, several years later, inspection of underwater photographs taken in July 2013 at Cayo Arcas reef, 350 km northwest of Coatzacoalcos, showed that it was present in significant numbers at both those sites at the same time in 2013 (Robertson et al., 2016a). Furthermore, this species was recorded in appreciable numbers on reefs near Veracruz, 200 km northwest from Coatzacoalcos and 450 km west from Cayo Arcas, in October 2014 (Robertson et al., 2016b). These observations indicate that N. cyanomos was already well established over a large area in the SwGoMx when it was first discovered, and indicate that its arrival in that area occurred long before 2013. More recently, N. cyanomos has been found to be even more widely spread across the SwGoMx (Schofield, 2017, accessed Dec. 20, 2017). Between 2016 and 2017, this species also was found at widely separated sites in the northern GoMx: on a natural reef off Texas in the west, on a petro-platform off Louisiana, and on an artificial reef off the Florida Panhandle in the east (Schofield, 2017, accessed Dec. 20, 2017).

The occurrence of N. cyanomos at two widely separated sites at the time it was first discovered in the SwGoMx demonstrates that the location at which any alien species is first discovered should not be equated with the site at which it was first introduced. This issue is particularly applicable to N. cyanomos, since it is small, dully colored, and readily mistaken for a native planktivorous damselfish, the brown chromis, Chromis multilineata (Guichenot, 1853), which it resembles in size, shape, and coloration. Distinguishing small juveniles of these two species in the field is more difficult than distinguishing adults. The brown chromis is particularly abundant on coral reefs in the SwGoMx, where it is far more numerous than N. cyanomos (D Robertson, pers. obs., 2016), and also lives on petro-platforms in the north and south of the GoMx. In that area both species often co-occur in feeding aggregations that are predominantly composed of brown chromis. Small numbers of N. cyanomos sharing a reef (or petro-platform) with many brown chromis could easily be ignored or mistaken for that species by a diver who is not familiar with, and not actively searching for N. cyanomos.

Given the difficulty in detecting the timing of the arrival of N. cyanomos in the SwGoMx, the possibility that it was initially introduced to the northern GoMx rather than the southwest part cannot be ruled out at present. Drilling-platforms originated from sites in the native geographic ranges of both lineages of N. cyanomos occur in both the northern and southern GoMx (Table 2). The fact that N. cyanomos seems to be much more common in the SwGoMx, even superabundant on petro-platforms there, and that it was first detected there, does suggest that it was initially introduced there. However, the greater abundance of this species in the SwGoMx may simply be due to more favourable environmental conditions (warmer water and greater availability of coral reefs), rather than prior occurrence there.

Future directions for research

Three sets of DNA barcode data from N. cyanomos at several sites in its native range should help clarify some of the questions concerning the origins of the SwGoMx population. First, as can be seen from the numbers of platforms dispatched to other countries from Singapore (http://www.infield.com/rigs), that city-state is a major global player in the business, has produced the largest number of GoMx petro-platforms that originated in the IWP, and is a site for servicing of used platforms coming from various areas, including the tropical Atlantic. Documenting which lineage(s) of N. cyanomos occur in Singapore would help evaluate the potential for Singapore as a source population for the Gulf of Mexico population of this damselfish. Second, a substantial set of DNA-sequence data also is needed from Gujarat, to indicate the proportional abundances of groups 1 and 4 lineages in the population there, and to show whether group 4 haplotypes present there are shared with west Pacific sites. Third, DNA barcode data from confirmed N. cyanomos from the Philippines would clarify the extent to which N. cyanomos is actually involved in the aquarium trade to the Americas, and whether Philippine fish belong to a DNA lineage found in the GoMx.

A host of other questions raised by the appearance of N. cyanomos in the GoMx deserve attention in future research. Prime examples include: (i) Where in the GoMx was N. cyanomos initially introduced: the northern or southern parts of that Gulf? (ii) Does N. cyanomos represent a true invasive species that will have negative effects on reef ecosystems of the Greater Caribbean, or a relatively innocuous addition to the reef-fish fauna of that region (cf. Davis et al., 2011)? (iii) What factors contribute to the evident success of this species in the Gulf of Mexico, on both petro-platforms and coral reefs? (iv) Why is N. cyanomos the only known successful alien reef-fish that appears to have arrived in the GoMx via relocated petro-platforms? Is that a reflection of the low frequency of wet-tows of rigs via southern Africa during intermittent warm-water episodes; or did other species arrive in numbers in the GoMx but fail to establish (cf. Oda & Parrish, 1981), or are there other alien fish species present there that have yet to be noticed (e.g., small, crevice-living, cryptic species)? (v) Has N. cyanomos, (or other IWP reef-fishes) established any unnoticed population(s) on platforms in offshore oil fields in other tropical parts of the Atlantic Ocean, such as western Africa and Brazil, which might have acted as stepping stones that helped this species spread into the GoMx on platforms or support vessels moved between Atlantic oil fields?

Conclusions

Various lines of evidence support the petro-platform translocation hypothesis: petro-platforms can provide habitat for substantial populations of a broad variety of tropical reef-fish; N. cyanomos can live in a dense population on a petro-platform; petro-platforms are regularly constructed within the parts of the native range of N. cyanomos where the SwGoMx lineages occur, and transported from there to the SwGoMx; translocated platforms has been implicated as vectors of long-distance transport of tropical reef-fishes, sometimes in sufficient numbers to allow them to establish breeding populations in non-native areas; members of the genus Neopomacentrus apparently can be transported long distances by bio-fouled structures in sufficient numbers to establish populations outside their native ranges; and, finally, the genetic diversity of the SwGoMx population of N. cyanomos indicates it is derived from a large and genetically diverse population of founders, consistent with large numbers of fish being transported by a drilling-platform. Neopomacentrus cyanomos has characteristics well suited to being transported by a petro-platform in sufficient numbers to establish a population far from its native geographic range: it is small (maximum length ∼12 cm), can feed on the plankton stream produced by a slow-moving platform (1–6 kts (2–11 kph); Yeo et al., 2010), and can live in high-density aggregations on small amounts of relatively low-profile, bio-fouling cover.

Heavy-lift ships, which carry offshore drilling-rigs on the deck, out of the water, often are employed for long-distance relocation of rigs. The use of this existing technology for all long-distance transport of such rigs would greatly reduce the risk of any types of marine organisms being transported outside their native ranges by relocated rigs. Such transport should be used not only for movements of rigs between oceans such as the Indo-Pacific and Atlantic, or the Mediterranean and Atlantic, but also relocations between regions with different biotas in the same ocean, such as those implicated in alien-fish transfers between the eastern and western tropical Atlantic.

Supplemental Information

Supplemental Information 1 Global shipping routes

Global Shipping Routes (from Halpern et al. 2008. A Global Map of Human Impact on Marine Ecosystems.Science 319: 948, DOI: 10.1126/science.1149345, Figure S2, with permission from AAAS). Red routes are the most heavily travelled.

Click here for additional data file.

Table S1 DNA Sequence data

Sample sites, sample sizes and data for DNA sequences of Neopomacentrus cyanomos and N. taeniurus referred to in this paper.

Click here for additional data file.

Appendix I Neopomacentrus taeniurus mtDNA sequences

Sequences of 3 confirmed Neopomacentrus taeniurus (Bleeker, 1856)

Click here for additional data file.

Tissue samples, sequences, and/or data for comparison were graciously provided by Gerald Allen and Mark Erdmann (Indonesia), Gavin Gouws, Roger Bills, and Phillip Heemstra (SW Indian Ocean), Luiz Rocha and Joseph Battista (Djibouti, Somaliland, and Saudi Arabia), Mark McGrouther and Jeff Leis (Australia), Nicolas Hubert (Madagascar), Dirk Steinke and Glenn Moore (Australia), Allan Connell (Mozambique), Fenton Walsh and Tim Bennett (Australia), Dhaval Bamaniya (India), Chiahao Chang and Kwang-Tsao Shao (Taiwan), Alyssa Marshell and Mark Priest (Oman), as well as Christopher Buerner and Adam Mangino of Quality Marine, Los Angeles, and Jason Edward of Greenwich Aquaria for aquarium trade specimens. Nicola Mulinaris of http://www.shipbreakingplatform.org supplied a list of petro-platforms, drill-ships and offshore support vessels sent to different shipbreaking sites during 2015–16. We thank Quetzalli Hernandez for help collecting specimens in the Gulf of Mexico, Rosa Gabriela Beltran López for support with the genetic analyses, and Maribel Badillo for photographs of the N. cyanomos specimens collected at Madagascar Reef. Giacomo Bernardi, Victor Seixas and Luis Malpica Cruz made useful comments of the manuscript.

Additional Information and Declarations

Competing Interests

Author Contributions

Animal Ethics

Field Study Permissions

DNA Deposition

D. Ross Robertson is an Academic Editor for PeerJ.

D. Ross Robertson conceived and designed the experiments, analyzed the data, wrote the paper, prepared figures and/or tables, reviewed drafts of the paper.

Omar Dominguez-Dominguez performed the experiments, analyzed the data, contributed reagents/materials/analysis tools, prepared figures and/or tables, reviewed drafts of the paper.

Benjamin Victor performed the experiments, analyzed the data, contributed reagents/materials/analysis tools, reviewed drafts of the paper.

Nuno Simoes contributed reagents/materials/analysis tools, reviewed drafts of the paper, sponsored fieldwork and collected specimens in the Gulf of Mexico used for genetic analyses.

The following information was supplied relating to ethical approvals (i.e., approving body and any reference numbers):

Collecting methods have been approved by the ACUC of the Smithsonian Tropical Research Institute (no. 2017-1107-2020).

The following information was supplied relating to field study approvals (i.e., approving body and any reference numbers):

Research at Cayo Arcas in 2016 was permitted by the Mexican Navy (SEMAR DO: 112/162), which also provided logistical support for that activity. Due to its proximity to the Mexican offshore oilfields, Cayo Arcas is a restricted area; access to it and research activities there are subject to approval by the Mexican Navy, which did so in this case and provided logistical support (transport) to the island.

The following information was supplied regarding the deposition of DNA sequences:

Genbank accession numbers of all sequences analysed are provided in Table S1.

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
