# Peer review of "An Indo-Pacific damselfish (Neopomacentrus cyanomos) in the Gulf of Mexico: origin and mode of introduction"

_PeerJ, doi:10.7717/peerj.4328_

## Round 0.1 · original submission · Minor Revisions

Both reviewers found your manuscript to be a solid contribution to invasive species research and recommended publication following changes largely related to clarity and completeness. I agree with this view and therefore recommend minor revisions.

The reviewers provide important suggestions regarding the completeness of the methods, alternative approaches to the analysis, and interpretation of the results. In addition, they provided helpful suggestions on the organization of the manuscript. I disagreed with some of the latter category and am therefore adding some comments below followed by a few minor suggestions not mentioned by the reviewers. You should consider my comments as alternative suggestions (like a third review) and follow whichever seem to make the most sense to you.

Many of the comments by Reviewer 1 were provided in an annotated pdf. When you revise your manuscript, you should list the substantive suggestions along with those in the formal review statement in your 'rebuttal' document. For the numerous suggestions regarding word use and grammar, don't bother with a specific response because I will be able to compare the reviewer annotations to your track changes document to make sure I agree with your decision or there is no misunderstanding. Please consider the wording suggestions carefully to see if your wording might have been confusing to a non-native English speaker. However, my view is that some of the proposed changes are not needed or incorrect.

With regard to the specific comments of Reviewer 1:
• I did not notice substantial problems with sentence construction.
• Some suggestions were changes from US to UK spelling. Note that PeerJ accepts either convention as long as the use is consistent throughout the manuscript.
• The reviewer objected to the use of initials identifying which author carried out specific aspects of the study such as the purchase or identifications. While I agree that all authors accept responsibility for all aspects of a manuscript to which their name is attached and it is not normal to indicate who did which part, I think it would not be a problem to use initials for the different researchers in a long-distance collaboration as this appears to be. (However, identifying who purchased the aquarium specimens is perhaps more detail than readers need!)
• The reviewer did not recognize the use of knots for ship speed. I suggest replacing knots with metric units. If you feel it is important to include the nautical term, provide the metric value in parentheses.
• I agree that capitalization is not needed for common names of species.
• I agree that adding 'see' to lists of references is superfluous.
• I agree that you do not need to reference the figures in the Methods.

Both reviewers suggested reorganization of the Introduction, so this should be considered seriously. However, I didn't have any difficulty with it on my first and second reads. I gave the topic some thought after reading the reviews, but was unable to arrive at a better proposal. Although the goals of the study come earlier than usual in the manuscript, I think it does make sense for the very specific focus of this study. The final paragraph of the Introduction outlining predictions from the three hypotheses seems appropriate for the Introduction. It may be possible, however, to consolidate the information on distribution of the species into the Introduction, as suggested by Reviewer 1, to reduce redundancy.

Additional editor comments:
L73 hypothesized (sp)
L84, 89, 96 delete 'see discussion' and present these points as assumptions that will be justified in the Discussion.
L107. You need a sub-head such as 'Study species and origin of specimens'
L183-187 of Results appear to be redundant with Methods
L192. Doesn't 'native to' refer to geographical area? As you are referring to habitat, would 'inhabits' be more appropriate?
L291. This is an awkward sentence. Revise to make the transit times from Hong Kong time and India parallel, rather than having time at the end of one phrase and the beginning of the next.
L306. choke point (two words)
L323, 342 West Africa (capitalize West)
L401, 441. Comma after 'Thus'
L447. city-state (hyphenated)
L523, 533-4, 580, 619. Inconsistent capital letters in journal article titles
Fig. 2, S2. Is there any need to credit a source for the map?
I note that you queried how to include collecting and animal care permits into the Methods. Because your statements on these topics seem appropriate, I will leave it to the journal staff to clarify where to insert the information.

·

Basic reporting

- The English should be improved, mainly in the introduction and methodology sections. More attention should be given to the sentence construction.
- The introduction and background is appropriated but can be improved in relation to the information about the species. A restructuring of the introduction is encouraged to make it clearer. The references used are in agreement with the subject addressed in the article, but more references should be added to reinforce your expected results.
- The structure are in conforms to PeerJ standards.
- The figures are relevant and the quality seems good. The labels should by improve and more detailed, mainly the Fig. 2 label.
- The sequences were deposited in GenBank and the accession numbers are provided.

Experimental design

- The research is in agreement with the PeerJ scope.
- The article question is relevant, meaningful and well-defined.
- The investigation was performed with an appropriate methodology.
- The methods description should be improved to be more understandable.

Validity of the findings

- The data is robust and statistically sound.
- The conclusions are well supported by the results obtained and linked to the original research question.

Additional comments

- Provide the species name in title “An Indo-Pacific damselfish (Neopomacentrus cyanomos) in the Gulf of Mexico: origin and mode of introduction.”
- In the M&M section I don’t understanding why you have two COI fragments, one of 594 bp and other of 652 bp. The “DNA sequence” part is confusing, which makes the understanding hard. Please consider reorganize the DNA sequence part.
- You not specify the program used to calculate the genetic diversity parameters.
- Provide the parameters used in the maximum likelihood analysis.
- The neutrality tests and mismatch distribution are nice to give an idea about the historical changes. However, a more sophisticate and less speculative analysis should be included. Thus, I suggest include a Bayesian Skyline Plot. You can do this using the program BEAST/BEAuTi.
- In the Result section, I recommend include a table with the genetic diversity parameters and the values of neutrality tests and mismatch distribution analysis. In addition, a separate figure with the ML tree should be nice (just a minor suggestion).
- Other minor comments are made in the attached document.

·

Basic reporting

Writing is clear overall and results in figures and tables well presented.

However I consider that the introduction could be structured better, I think authors jump too quickly to present their research questions and the last paragraph of introduction reads a bit like a discussion, it would be useful to tidy it up a bit to make it more concrete and short.

The last paragraph in discussion section/subsection “Where in the GoMx was N. cyanomos initially introduced?”, lines 446-457, is very important as future directions, however it does not fit with the overall theme of this subsection, I would suggest change the name of subsection, distribute its contents along the discussion of each of the three hypothesis, or add a new subsection.

Conclusions section, lines 468-469, authors refer to the Neopomacentrus species, perhaps should be the Neopomacentrus genus?

Figures overall, it would help in the interpretation if the same color palette between Figure 1 and 2 is used as currently different blues are used for groups 1 and 4.

Figure 1. Please add a clear description of what each slice of pie represents.

Experimental design

Research questions are well defined and analysis is implemented thoroughly, overall I do not see any flaws on this area. However I do think that in the Methods section (particularly lines 121-124), it is not very clear which data was collected for this study and which data comes either from other studies or collected for other purposes.

Validity of the findings

I consider the overall interpretation of data solid, however I am missing a bit more discussion to strengthen the discussion presented on the competing hypothesis, in particular:

I understand that the molecular markers used do not support the first hypothesis, however I think authors are dismissing it very rapidly without digging it out completely, a couple important points: 1) stating that marine aquarium trade is negligible in Mexico based on “personal communication” (line 257) is very weak to say the least, perhaps cite: Lango-Reynoso et al 2012 (Lat. Am. J. Aquat. Res., 40(1): 12-21, 2012. DOI: 10.3856/vol40-issue1-fulltext-2) or other paper that actually proves or show this is the case; 2) additionally, I think the idea in line 455-457 is kind of lost and out of place here as this is an important point to make in light of data currently analyzed and remaining research avenues to explore.

I find that authors did a solid job at discussing why hypothesis 2 is not plausible in terms of the most used shipping routes for cargo ships, however I have some concerns and fail to see the same thorough link between the ecological plausibility of a fish species such as N. cyanomos surviving a journey from the Indo-Pacific to the SwGoMx in petro platforms, that is when discussing hypothesis 3. In particular, while authors do present examples from other introductions, all these have been comparatively less traveled distances from origin to final destination. Additionally, supplemental Figure 2 indicates the origin of platforms as a set of arrows from the origin to the final destination, however it is not clear to me whether this is a depiction of their actual traveling paths? If so, tropical sessile fouling organisms and associated fish would need to endure a travel through temperate water areas in South Africa as cold as those present in California (annual mean temperature range for both location is ~14-22 C), which is one point authors present against hypothesis 1 (lines 297-299). Authors should address these concern or specify the likely travel path of the petro-platforms.

Furthermore, and while I understand this may not be the goal of the manuscript, I think authors should present or state as future research directions why – if these petro platforms/drilling ships carry such large communities – have we failed to see more invasive species establishing in the SwGoMx, has it been a lack of detectability, or perhaps some specific characteristics inherent to the studied species N. cyanomos?

Additional comments

This paper is very relevant for the field of invasive species in coral reef ecosystems as there is a need to understand the vectors of invasive species to implement control and management strategies as to minimize such introductions. This paper is also relevant in light of recent invasive species that have produced large and potentially everlasting ecological damage to invaded areas such as that of the invasive Indo-Pacific lionfish to the North Western Atlantic. Authors did a great job at presenting solid research questions and addressed them accordingly.

---

## Round 0.2 · accepted · Accept

Appropriate changes have been carried out and the manuscript is now suitable for publication.

I have a few minor corrections.

The authors disagreed with suggestion to capitalize West Africa because it is not a country. However, as a defined region, it may still need capitalization (as, for example, North America or West Africa on the Wikipedia page). Production staff should decide. L368, 385 (OK in references, Table 2)

A few commas missing:
Thus L229, 338
In addition L142, 337